# Chemokine/Interleukin Imbalance Aggravates the Pathology of Respiratory Syncytial Virus Infection

**DOI:** 10.3390/jcm11206042

**Published:** 2022-10-13

**Authors:** Kentaro Mori, Takeaki Sasamoto, Tetsuo Nakayama, Shinichiro Morichi, Yasuyo Kashiwagi, Akihito Sawada, Hisashi Kawashima

**Affiliations:** 1Department of Pediatrics and Adolescent Medicine, Tokyo Medical University, 6-7-1 Nishishinjuku, Shinjuku-ku, Tokyo 160-0023, Japan; 2Department of Viral Infection I, Omura Satoshi Memorial Institute, Tokyo 108-8641, Japan

**Keywords:** CX3CL1, fractalkine, CCL21, IP-10, IL-6, severity, imbalance

## Abstract

(1) Background: Almost 100% of children are initially infected by respiratory syncytial virus (RSV) by the age of 2 years, with 30% to 40% of children developing lower respiratory tract infections, of which 1% to 3% become severe. The severity of RSV-induced disease correlates with the influx of leukocytes, which leads to damage of the airways. We hence performed an immunological study based on the assumption that a chemokine/interleukin imbalance affects respiratory disorders caused by bronchiolitis and severe pneumonia. (2) Methods: The subjects were 19 infants without any underlying diseases, who developed respiratory symptoms owing to RSV infection. The subjects were stratified by their symptom severity, and chemokine and interleukin levels in their serum and tracheal aspirate fluid (TAF) were measured. (3) Results: The data of TAF, which were only obtained from subjects with severe symptoms, indicated that levels of inflammatory interleukins were much lower than the levels of chemokines. Three out of 6 subjects with severe symptoms showed below detectable levels of IL-6. TNF-α and IFN-γ levels were also lower than those of chemokines. The main increased CCL chemokines were CCL21 and CCL25, and the main increased CXCL chemokines were CXCL5, 8, 10, 12, and CX3CL1 in the lower respiratory region. Multiple regression analysis demonstrated that serum CX3CL1 and IL-6 levels were most strongly associated with symptom severity. This is the first report to date demonstrating that serum CX3CL1 level is associated with the severity of RSV infection. (4) Conclusions: Our results demonstrated that specific chemokines and the imbalance of cytokines are suspected to be associated with aggravated symptoms of RSV infection.

## 1. Introduction

Almost 100% of children are initially infected with respiratory syncytial virus (RSV), a non-segmented, single-stranded RNA virus, by the age of 2 years, of whom 30% to 40% develop lower respiratory tract infections, and 1% to 3% develop severe symptoms [1]. In addition, infant death caused by RSV infection is not rare [2]. RSV-associated deaths among children younger than 5 years of age are thought to be common, estimated at 100 to 500 per year, which is more than the number of infant deaths caused by influenza in the United States [3]. The severity of RSV-induced disease correlates with the influx of leukocytes, which leads to damage of the airways [4]. Chemokines are the main cytokines that recruit leukocytes into the airways. Chemokines/interleukins are thought to be involved in the damage resulting from respiratory disorders caused by RSV infection, as well as in the recovery from these diseases. However, the association between the balance of chemokines/interleukins and disease severity has remained unclear to date. In this study we performed immunological analyses of various chemokines and interleukins in the airways of RSV-infected patients.

## 2. Material and Methods

The subjects were 19 infants with no underlying diseases, who developed respiratory symptoms owing to RSV infection. The subjects comprised 9 boys and 10 girls, aged from 1 month to 2 years. The nasal fluid samples tested using a rapid assay for RSV (Check RSV; Alfresa, Japan) were positive in all subjects. The breakdown was death/requiring mechanical ventilation in 6 subjects, requiring oxygen therapy in 6 subjects, and not requiring oxygen therapy in 9 subjects. According to their level of oxygen demand, the patients were divided into 3 groups. The serum samples were obtained with informed consent from the parents at the time of admission. Tracheal aspirate fluid (TAF) was obtained from patients who required mechanical ventilation, and was subjected without dilution. The subjects were stratified by severity as above, and 48 chemokines/interleukins in the serum and TAF were measured by Bio-Plex Multiplex Cytokine Assay (Bio-Plex Pro^TM^ Human Chemokine Panel in Bio-Rad Laboratories, Inc., Hercules, CA, USA). The analyzed chemokines/interleukins were CCL-1, -2, -3, -7, -8, -11, -13, -17, -19, -20, -21, -22, -23, -24, -25, -26, -27, CX3CL1, CXCL-1, -2, -5, -6, -8, -9, -10, -11, -12, -13, -16, granulocyte macrophage colony-stimulating factor, interferon (IFN)-γ, interleukin (IL)-1β, -2, -4, -6, -10, -16, monocyte chemoattractant protein (MCP)-1, -2, -3, macrophage migration inhibitory factor (MIF), and tumor necrosis factor (TNF)-α. Statistical analyses were performed using IBM^®^ SPSS^®^ Statistics version 25.0 software (New York, NY, USA). A Pearson *r*-coefficient of greater than 0.8 was considered to indicate a statistically significant correlation (*p*-value < 0.05). After hierarchization of the patients by symptom severity, statistical analyses were performed. Serum from 10 healthy controls (from infants to adults) and 4 patients with underlying diseases and RSV infection were also analyzed.

## 3. Results

From the data of TAF (Figure 1), which were obtained only from subjects with severe symptoms, the levels of interleukins were much lower than those of chemokines between subjected interleukins and chemokines quantitatively. The median levels and standard deviations (SD) of IL-1β, -2, -4, -6, -10, and -16 in the acute phase were 24.46 (276.62), below the detection level (0), below the detection level (8.16), 14.18 (16.8), 22.73 (14.77), and 10.0 (246.8) pg/mL, respectively. Three out of 6 patients showed below detection levels of IL-6. The level of MIF in TAF was the highest among the tested molecules (mean ± SD; 56,932 ± 71,960 pg/mL). TNF-α and IFN-γ levels were detectable (mean ± SD; 42 ± 31.7 pg/mL and 66 ± 81.3 pg/mL, respectively). Among the detectable chemokines, CCL7, CCL11, CCL19, CCL20, and CCL27 showed no fluctuations between the acute phase and convalescent phase of disease, as shown in Figure 2. The TAF data demonstrated that the main CCL chemokines expressed during the disease phase were CCL21 and CCL25, and the main CXCL chemokines expressed were CXCL5, 8, 10, 12, and CX3CL1, and no interleukins were expressed at substantial levels (Figure 3).

Serum data from all children, including those with mild and severe symptoms, showed that the levels of CX3CL1, IL-6, CXCL12, and CXCL16 significantly correlated with the severity of RSV infection (Figure 4). On multiple regression analysis, CX3CL1 and IL-6 were found to be serum factors with the strongest association with symptom severity. Serum levels of CX3CL1 significantly correlated with those of IFN-γ, IL-10, and CXCL10 (IP-10), as shown in Figure 5.

**Figure 1 jcm-11-06042-f001:**
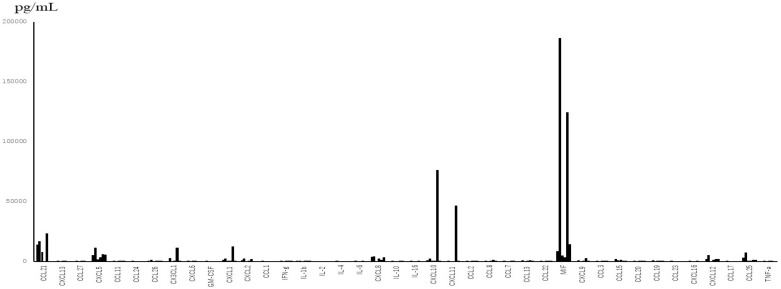
Levels of 48 chemokines and interleukins in the tracheal aspirate fluid (TAF) obtained from 6 patients with severe disease during the acute phase. The main detectable cytokines in TAF were chemokines rather than inflammatory cytokines (interleukins).

**Figure 2 jcm-11-06042-f002:**
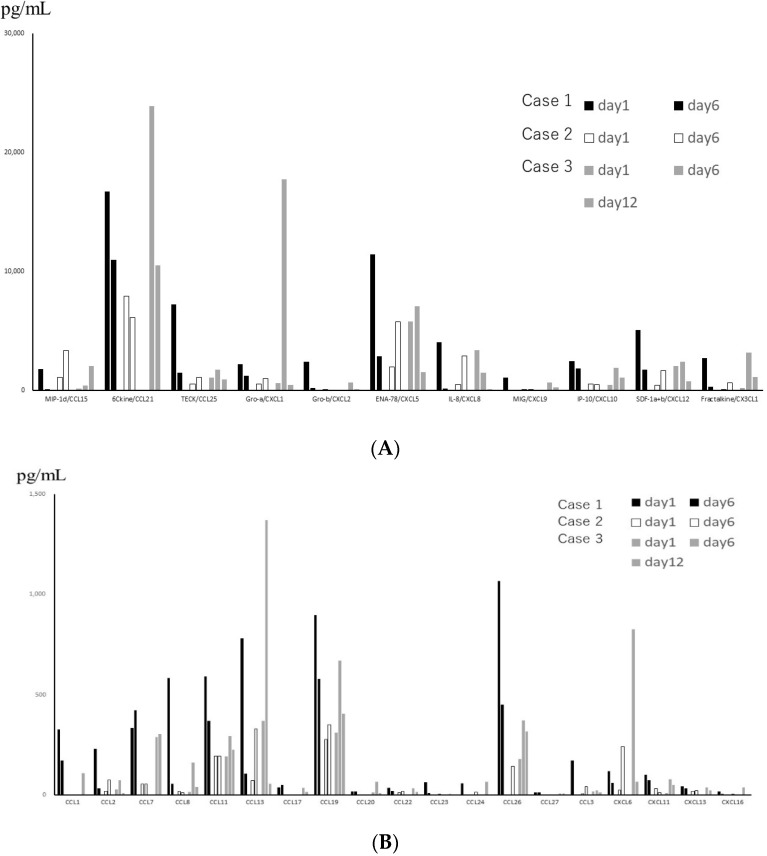
(**A**) Levels of chemokines with high expression in Tracheal aspirate fluid (TAF) obtained from 3 patients with severe disease, on different days. Black, white and gray bars represent the data of 3 different patients. (**B**) Levels of chemokines with low expression in TAF obtained from 3 patients with severe disease, on different days.

**Figure 3 jcm-11-06042-f003:**
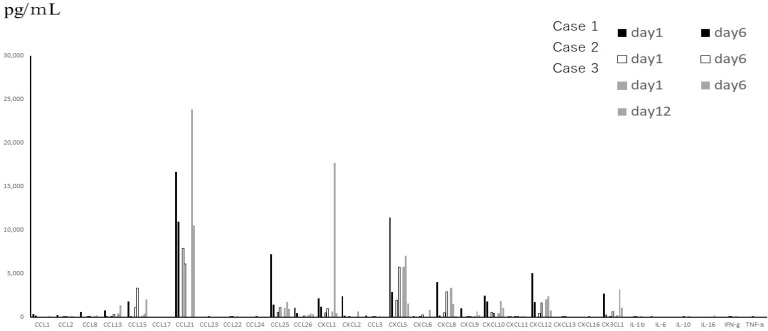
Main cytokines, including inflammatory cytokines expressed in TAF, which were obtained from 3 patients with severe disease, on different days.

**Figure 4 jcm-11-06042-f004:**
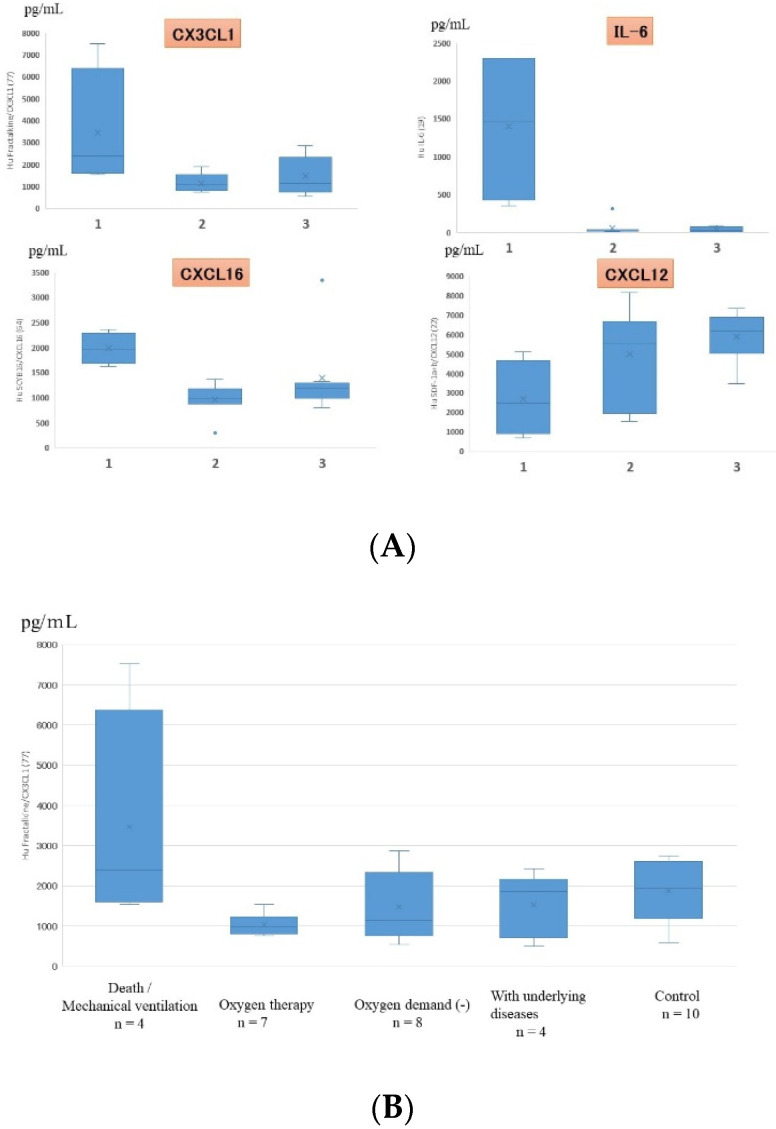
(**A**) Cytokines that showed statistically significant differences in serum expression levels in children with different severities of RSV infection (only 4 out of the 48 cytokines showed statistically significant differences). 1: death/mechanical ventilation; 2: oxygen therapy; 3: oxygen demand (-). (**B**) Serum fractalkine/CX3CL1 levels in patients with different severities of RSV infection. Box and whisker plot (box: interquartile range; whiskers: minimum and maximum).

**Figure 5 jcm-11-06042-f005:**
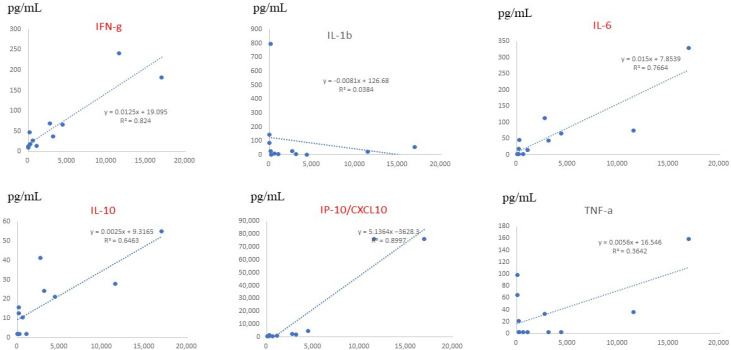
Comparison between serum fractalkine/CX3CL1 levels and inflammatory cytokine levels in patients with RSV infection. The horizontal lines (*x*-axis) indicate the level of each cytokine, and the vertical lines (*y*-axis) indicate CX3CL1 levels. *R* represents the Pearson *R*-correlation. IFN-γ, IL-6, IL-10, and CXCL10 showed statistically significant differences between each cytokine and fractalkine/CX3CL1.

## 4. Discussion

Clinical studies have suggested that the severity of RSV-induced disease correlates with the influx of leukocytes, which leads to damage of the airways [5]. Chemokine levels have been shown to correlate directly with the intensity of inflammatory responses, and chemokine expression can be induced by viral infection in multiple pulmonary cell populations, including neutrophils. Neutrophils are the dominant inflammatory cell in the airway of children with bronchiolitis caused by RSV infection [6,7]. It is not known, however, whether individuals who are more susceptible to severe RSV infections have different profiles of cytokines, particularly chemokines, compared with those who are less susceptible.

We previously reported that chemokines play roles in the neurological complications that occur in RSV-infected children. In a study profiling 17 cytokines from the cerebrospinal fluid of 8 RSV-infected children with neurological complications, 1 patient was found to have high levels of 13 cytokines and was considered to be experiencing a cytokine storm, and the other 7 patients were found to have high levels of IL-6, CXCL8, CCL2, and CCL4 [8].

An investigation in Japan reported that the causes for RSV-associated hospitalizations in 359 children without underlying diseases were severe bronchiolitis (81.6%), RSV encephalitis (5.3%), near-miss sudden infant death syndrome (SIDS) (3.1%), RSV myocarditis (1.4%), SIDS (0.3%), and others (8.4%) [9]. This demonstrates that patients with severe symptoms most frequently have respiratory infections, including serious bronchiolitis. In the present study, we investigated the levels of cytokines, particularly chemokines in the serum and TAF of children with RSV infection. TAF samples were obtained from children who had severe respiratory symptoms. The data of TAF demonstrated that the levels of interleukins were much lower than those of chemokines (approximately less than 1/100). The main CCL chemokines expressed in the TAF of the patients with severe symptoms were CCL21 and CCL25, and the main CXCL chemokines were CXCL5, CXCL8, CXCL10, CXCL12, and CX3CL1, and no interleukins, such as inflammatory cytokines, were expressed at substantial levels. RSV can directly induce the expression of inflammatory cytokines and chemokines from both airway epithelial cells and resident macrophage populations, such as TNF, IL-l, IL-6, and CXCL8 (IL-8) [10,11]. These results demonstrate that the lack of several inflammatory cytokine responses might cause severe infection in the lower respiratory tract. Moreover, other viruses, i.e., influenza virus, rhinovirus, adenovirus, etc., might induce chemokine production in the TAF.

In this study, we analyzed serum chemokine and interleukin levels in patients with RSV infection and controls. There were substantial differences in the levels of inflammatory and anti-inflammatory cytokines between the patients and controls. A number of studies on chemokines demonstrated that high levels of CCL3 (MIP-1α), CXCL8, CCL2 (MCP-1), and CCL5 (RANTES) released during RSV infection are associated with severe RSV infections in infants [12,13,14]. Garofalo et al. also reported that nasopharyngeal secretions samples from infants with severe RSV infections have a high level of chemokines (CCL2 and CCL3), and hence these chemokines are associated with disease severity [15]. An in vitro study demonstrated that RSV-infected epithelial cells appear to be a rich source of a number of different chemokines, including CCL2, CCL3, CCL5, and CXCL8, which are induced via the activation of NFĸB [16]. Several of these chemokines, including CCL3 and CCL5 in the airway, have also been shown to be associated with inflammatory responses in animal models of RSV infection [17]. Previous reports concluded that CCL3 level is associated with the severity of primary RSV-induced inflammation as well as with repeated RSV infections. In addition, studies analyzing CCL5 have indicated its significant effects on the pathophysiological responses of primary RSV infection as well as the pattern of leukocyte recruitment and leukotriene release in the lungs during RSV-induced allergen-exacerbated disease. The overproduction of mucus and the development of airway hyperreactivity in mice were shown to be directly associated with the expression and activation of CXCR2 (an IL-8 receptor homolog), which is a receptor associated with neutrophilia [18]. CXCLl0 (IP-10) has been shown to be associated with RSV clearance [19]. In the present study, we found that serum CX3CL1 level, but not CCL3 level is associated with the severity of RSV symptoms. We suspect that the association with CCL3 and CCL2 levels did not reach statistical significance because of the small number of subjects and the sampling method of the TAF used.

CX3C chemokines have the characteristic in which there are 3 amino acids between the 2 cysteine residues on the N-terminal side. Fractalkine/CX3CL1 and CXCL16 are called membrane-binding chemokines, and have roles as chemotaxic adhesion molecules, which differ from other CC and CXC chemokines. It has been suggested that that immunological responses are partially modified through the interaction of viral G-glycoprotein with the host’s chemokine receptor CX3CR1, and a polymorphism in CX3CR was associated with the severity of RSV bronchitis [20]. The natural ligand of CX3CL1 (fractalkine) also affected the G glycoprotein-CX3CR1 pathway. The chemokine CX3CL1 has a similar structure to the G protein of RSV. As in the G protein, CX3CL1 has a CX3C motif that is involved in forming dual disulfide bonds [21]. CX3CL1 binds via the CX3C motif to the CX3CR1 on macrophages/monocytes, dendritic cells, lymphocytes, natural killer cells, and epithelial cells [22,23]. The membrane-bound form of CX3CL1 is expressed on activated epithelial cells and binds to CX3CR1 on inflammatory cells [24,25]. After cleavage by a metalloproteinase, its ectodomain is released as the secreted form of CX3CL1. Secreted CX3CL1 acts as a chemoattractant for inflammatory cells that express CX3CR1 [24,26]. In our present study, serum CX3CL1 level associated with the severity of RSV infection in patients. CX3CL1 levels statistically correlated with IFN-γ, IL-6, IL-10, and CXCL10 (IP-10) levels in blood. Mice infected with RSV after the administration of CXCL10 presented with more fulminant and necrotizing diffuse alveolar and bronchiolar damage with lymphocyte infiltration [27]. Therefore, CX3CL1 is a pivotal circulating chemokine for determining the severity of RSV infections.

CX3CL1 is a unique chemokine comprised of 373 amino acids, consisting of an extracellular N-terminal domain, a mucin-like stalk, a transmembrane alpha helix, and a short cytoplasmic tail. The soluble form of CX3CL1 that we detected in this study induces chemotactic activity in monocytes, NK cells, and T cells. CX3CL1 induced by TNF-alpha, IFN-gamma, and IL-1-beta play roles in various diseases and complications, such as atherosclerosis, rheumatoid arthritis (RA), HIV infection, cancer, etc. Serum CX3CL1 levels are increased in patients with RA and patients with SLE, and correlate disease activity [28,29]. Moreover, serum levels of CX3CL1 were found to be high in RA patients with rheumatoid vasculitis, and to correlate with vasculitis disease severity and levels of inflammatory markers [30]. In accordance with these previous findings, high levels of CX3CL1 in patients with severe RSV infection in this study might be correlated with vasculitis.

On the other hand, CX3CR1 is also expressed and secreted by neurons and reduces the expression of proinflammatory genes [31]. CSF (cerebrospinal fluid) and serum fractalkine levels were also found to be significantly increased in multiple sclerosis patients [32]. Serum concentrations of fractalkine have been reported to be significantly higher in HIV-infected patients with CNS complications than HIV-positive patients without CNS complications, and compared with HIV-negative controls [33]. Severe RSV infection has been reported to cause central nervous system such as encephalopathy, SIDH (Syndrome of inappropriate secretion of antidiuretic hormone) and others [34]. Therefore, latent CNS lesions may be involved in severe RSV infections and sudden death of RSV-infected patients.

Taken together, the profile of chemokine production indicates whether immune cell recruitment will lead to severe disease, efficient clearance of the virus, and/or subsequent development of chronic pulmonary disease. In this study, severe disease was dominantly observed in very young patients. Therefore, the results may change if the age distribution is different among the groups. Although no specific clinical treatment for RSV infection is currently available, common drugs and new antagonists that target fractalkine/CX3CL1 might be developed as new treatments for RSV infection in the future.

The limitations of this study are the number of samples, particularly of the TAF. TAF from controls and patients with mild disease could not be obtained because of invasive procedure. It has been reported that bacterial infections were observed in approximately 40% of children with RSV infection [35]. In addition, *Streptococcus pneumoniae* was shown to enhance RSV infections both in vitro and in vivo [36]. Therefore, differences in disease severity may be a result of the coinfection of bacteria. To obtain more insight into the functions of CX3CL1, additional research using animal models of RSV infection is needed in the future.

## 5. Conclusions

Specific chemokines are important for the pathophysiology of RSV infection, and new treatments targeting specific chemokines are expected to be developed as effective therapies in the future.

## Data Availability

Not applicable.

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
