# Peer review of "Chemokine/Interleukin Imbalance Aggravates the Pathology of Respiratory Syncytial Virus Infection"

_jcm, 2022, doi:10.3390/jcm11206042_

Round 1

Reviewer 1 Report

The finding that serum CX3CL1 level is associated with the severity of RSV infection is quite interesting. However, the study has several weaknesses:

1. The figures need to be modified, since they are not readable, even when highly magnified in PDF format. Instead to be horizontal, they could be presented vertically, so the names of the chemokines could be bigger.

2. The sample size is quite small, especially in case of TAF study (only 3 samples tested), no data presented from the control groups. As immuno-modulator, the airway epithelial cells express and secrete a large amount of cytokines and chemokines, even at normal health conditions, usually reaching ng/ml range. Therefore, the fold of increase observed by the authors is rather mpdest and insignificant. 

3. The only significant increase of CX3CL1 in the serum was observed in the group of "Death". Therefore, one could argue that this increase might be the consequence of death, rather than a direct effect of RSV infection. By the way, there was no measure of viral load in this study. The viral charge alone might explain the severity of the diseases. 

3. Besides RSV infection, the children might be over-infected by other viruses or bacteria. Double infections are common phenomenon in children. Thus, the severity of the disease may be caused by double infection. This possibility has not be considered.

4. The main finding is that serum CX3CL1 level is associated with the severity of RSV infection. Thus, the authors should put more emphasis on CX3CL1, by citing more references about the mechanism and its implication in various diseases.

As conclusion, CX3CL1 is indeed an interesting and important chemokine in various diseases. But this article failed to provide more insight about CX3CL1 function.  

Reviewer 2 Report

Dear Editor,

Thank you so much for sending me this manuscript. I found the study exhaustive and very well organized. This work helps in the understanding of specific chemokines and the imbalance of cytokines that are associated with aggravated symptoms of RSV infection. The objective is clearly stated and fulfilled. The quality of the English is satisfactory, and also the article is presented very well. However, I suggest adding the study's limitations at the end of the discussion. And the authors should improve the Introduction.

Kind Regards,

Reviewer 3 Report

Dear Editor,

the manuscript entitled Chemokine/interleukin imbalance aggravates the pathology of respiratory syncytial virus infection is dealing with actual problem regarding severe RSV infections and what may cause this.

The main remarks regarding the study are that it was done on limited number of subjects and that authors didn't discuss this limitation. Therefore, my suggestion is that discussion and conclusion rephrase in terms on study limitation and that conclusions are not so strait forward.

 It would be better to discuss study limitations in context of results. This would be appreciated and the paper could be more appropriate for publication. These changes should be in the manuscript and in the abstract as well.

Best regards!

Round 2

Reviewer 3 Report

Dear Editor,

this manuscript deals with respiratory syncytial virus (RSV) that causes a major burden in public health, in developing as well as in industrialized countries. RSV infections may affect lower respiratory system and cause severe symptoms, especially in infants. Therefore, it is important to understand why this is happening and can we do anything to detect high risk patients.

Several issues were raised during the review process in order to improve the manuscript and authors answered all issues. There are no further issues regarding this manuscript.

Best regards!